# Minimally Invasive Approaching in Hip Surgery—An Anatomical Investigation of 20 Specimens

**DOI:** 10.3390/medicina57111283

**Published:** 2021-11-22

**Authors:** Clemens Schopper, Hannes Traxler, Bernhard Schauer, Günter Hipmair, Tobias Gotterbarm, Matthias Luger

**Affiliations:** 1Department for Orthopaedics and Traumatology, Kepler University Hospital GmbH, Johannes Kepler University Linz, Krankenhausstrasse 9, 4020 Linz and Altenberger Strasse 69, 4040 Linz, Austria; bernhard.schauer@kepleruniklinikum.at (B.S.); guenter.hipmair@kepleruniklinikum.at (G.H.); tobias.gotterbarm@kepleruniklinikum.at (T.G.); Matthias.luger@kepleruniklinikum.at (M.L.); 2Centre for Anatomy and Cellbiology, Department of Anatomy, Medical University Vienna, Waehringerstrasse 13, 1090 Vienna, Austria; hannes.traxler@meduniwien.ac.at

**Keywords:** hip surgery, minimal invasive approach, anterior approach, Lateral Femoral Cutaneous Nerve, hip replacement, cervical neck fracture

## Abstract

*Background and objectives*: Based on the preparation of 20 formalin-fixed anatomical cadavers, the feasibility of the anterior, minimally invasive approach to the hip joint was investigated in each side of the body. The hypothesis of the study was that the Lateral Femoral Cutaneous Nerve can be spared under the use of this approach. *Materials and Methods*: The anterior approach to the hip was performed via an incision of 8 cm. The position of the nerve was noticed in relation to the skin incision, and the distance was measured in millimeters. The nerves main, gluteal and femoral trunk were distinguished and investigated for injury. *Results*: No injury of the main trunk was noticed. The average distance of the main trunk to the skin incision was 14.9 and 15.05 mm in the medial direction, respectively (*p* < 0.001). Injury of the gluteal branch has to be considered at an overall rate of 40%. *Conclusions*: The anterior, minimally invasive approach to the hip joint can be performed without injury of the Lateral Femoral Cutaneous Nerve.

## 1. Introduction

Performance of a minimally invasive styled approach to the hip joint has become a state-of-the-art-procedure in hip replacement surgery throughout the past two decades [1,2,3,4,5,6]. Bullet points like less intra- and postoperative blood loss, faster physical recovering, shorter hospitalization and shorter surgical intervention time are clear advantages that have made minimally invasive approaches a highly accepted procedure [7,8,9,10,11]. A popular representative of this kind is the anterior approach to the hip joint. The approach is performed through a muscle sparing interval between the tensor fasciae latae and the sartorius muscle [12,13]. This approach was mentioned in its first variant by Sutherland et al. in 1944, and was described first in its recent form by Light et al. in 1980 [13,14]. It offers the following advantages: (1) The hip joint can be reached in the shortest way [15], (2) The approach is performed in an area which is known for its sporadic innervation [16,17], and (3) performance of this approach is possible without harming a single muscle or tendon [12]. In contrast to these advantages, the performance of this approach also brings disadvantages, such as a flat learning curve and a reduced overview over the operation situs for the performing surgeon. Furthermore, the topographic nearness of the large inguinal vessels and the Lateral Femoral Cutaneous Nerve (LFCN) has to be respected as well [15,18,19,20]. Injury of this nerve can lead to a pain syndrome known as Meralgia Paraesthetica, causing disturbing sensations and pain in the area of the lateral thigh [21]. Amongst others, the existing data concerning the issue of the LFCN and its injury during approach to the hip joint refers to a clinically retrospective work from Bhargava et al. and an anatomic work from Ropars et al. [9,22]. The first reports an incidence of LFCN injury at 14.8% incidence identifying 12 out of 81 patients with paresthesias after performance of a hip arthroplasty through an anterior approach. The authors of the latter designed a map of danger zones according to their anatomic findings regarding the LFCN. This map was distinguished into subdivisions separating the LFCN and its following branches, the femoral and gluteal as well as the risk of injury during performance of the anterior approach to the hip joint. Three zones were defined: the main trunk, gluteal trunk and femoral trunk. In conclusion, the authors recommend a skin incision as lateral and as distal as possible to the anterior iliac spine (ASIS) to avoid injury of the LFCN [22]. The aim of the current study was to define the position and injury of the LFCN’s main trunk according to the skin incision performed under use of an anterior approach to the hip joint. We hypothesized that the anterior approach to the hip joint is performable without injury to the LFCN’s main trunk.

## 2. Materials and Methods

This work was approved by the local ethics committee (EK nr 1199/2011). Based on the dissection of 20 formaldehyde fixed cadavers (13 females/seven males) the skin incision of the anterior approach to the hip joint was performed on each side of the body. Including criteria were intact skin- and soft- tissue conditions and no preexisting operative intervention in the area of dissection. Following the example set by Paillard et al., the position of the incision was located in relation to a point set 20 mm dorsally and 10 mm distally to the ASIS as the proximal starting point and the caput fibulae as the distal ending point of the skin incision (Figure 1) [23]. The points were marked with a pin (Figure 2). The axis of the skin incision in between those two points was defined by tensing a yarn from one pin to the other. Following the line set by these two anatomical landmarks, the skin incision was performed over a length of eight cm by incising strongly through the skin, subcutis and the Scarpa fascia. The nerve’s main trunk, usually located in a fat-filled space between Scarpa Fascia and the fascia lata was identified. After identification of the nerve, its integrity was checked. The location in respect to the approach (medial/lateral) was noticed and the distance of the nerve’s main trunk was measured on a normal axis to the skin incision; results were scaled in millimeters. Furthermore, the nerve’s following branches, the gluteal and the femoral trunk, were searched and checked for integrity. Finally, the point of crossing between the gluteal trunk and the line of the skin incision was evaluated referring to its distance from the starting point of the incision. This point referred to the starting point of the skin incision, 20 mm dorsal and 10 mm distal to the ASIS. The main parameters consisting of the position of the nerve’s main trunk to the incision, its integrity and its distance to the line of incision were checked as well as the side- and intersexual difference. Statistical analysis of the parameters of interest was performed with a statistical software package (IBM SPSS Statistics, V23, IBM, Armonk, NY, USA). A Shapiro-Wilk test was conducted to screen the data for normality of distribution, and a Welch test was performed to compare the gender-related results. A G-power analysis displayed a power of 80% under the use of this sample size.

## 3. Results

The location of the LFCN was recognized consistently in the fat-filled space between the Scarpa Fascia and the fascia lata. No injury of the nerve’s main trunk was recognized. Findings concerning the position of the nerve’s main trunk on the left side of the body showed the following distribution: In 20 of 20 cases (100%) the main trunk was located in medial direction to the skin incision. The mean distance of the main trunk was measured at 14.9 ± 1.19 mm (CI 95%, [12.42–17.38]) in medial direction to the skin incision (*p* < 0.001, Figure 3, Table 1). Findings concerning the position of the nerve’s main trunk on the right side of the body showed the following distribution: In 20 of 20 cases (100%) the main trunk was located in medial direction to the skin incision. The mean distance of the main trunk was measured at 15.05 ± 1.53 mm (CI 95%, [11.86; 18.24]) in medial direction to the skin incision (*p* < 0.001, Figure 4, Table 2). Findings concerning gender-related differences of the nerve’s distance to the skin incision did not show a significant difference (14.36 mm in medial direction in the male group, 15.31 mm in medial direction to the incision in the female group (CI 95%, [−5.3; 3.4]) (*p* < 1) Findings concerning the position and the crossing point of the gluteal branch with the line of skin incision on the left side of the body showed the following distribution: In 20 of 20 cases (100%) the gluteal branch was identified, in 10 of 20 cases (50%) the branch was injured by incising (Figure 5). In 15 cases (75%) the gluteal branch crossed the line of skin incision distal from the ASIS, in five (25%) cases it crossed the proximal. In 10 of 15 cases the branch crossed distal from the ASIS and was located in the area of the skin incision. The mean distance from the crossing point to the ASIS was 28.4 ± 8.62 mm (CI 95%, [46.44–10.36]) (*p* < 0.05). Findings concerning the position of the crossing point of the gluteal branch with the line of skin incision on the right side of the body showed the following distribution: In 20 of 20 cases (100%) the gluteal branch was identified, in six of 20 cases (30%) the branch got injured by incising. In 12 cases (60%) the gluteal branch crossed the line of skin incision distal from the ASIS, in eight (40%) cases it crossed proximal. In six of 12 cases the branch crossed distal from the ASIS it was located in the area of the skin incision. The mean distance from the crossing point to the ASIS was 36.15 ± 12.45 mm (CI 95%, [62.20–10.10]) (*p* < 0.05). Findings concerning gender-related differences of the crossing point of the gluteal branch with the line of skin incision showed the following distribution: in the male group the mean distance was found at 32.71 mm distal to the starting point of the skin incision. In the female group the mean distance was found at 32.04 mm distal to the starting point. A significant difference between the groups was not evident (CI 95%, [−38.2; 36.85]) (*p* < 1). Negative values indicate a crossing point distal to the starting point of the skin incision, while positive values indicate a proximal one. The femoral branch did not cross the line of skin incision in any of the investigated cases.

## 4. Discussion

In the course of this anatomical preparation study the position and injury likelihood of the LFCN were evaluated in regard to the performance of the minimal invasive anterior approach to the hip joint. It was verified that the performance of this approach is possible without injury to the LFCN’s main trunk. The approach was defined by the axis between the ASIS and caput fibulae. The skin incision was drawn laterally and dorsally to this axis. Another variant of this approach localizes the skin incision more medially between the tensor fasciae latae muscle and the sartorius muscle. Ropars et al. investigated the crossing point between the gluteal/femoral branch and the ventral edge of the tensor fasciae latae muscle. The femoral branch was found in between the tensor fasciae latae muscle and the sartorius muscle in 53% of the investigated cases. As a consequence, the femoral branch is at risk of injury under the performance of this approach, although it is performable without harming any muscle or tendon [22]. The femoral branch didn’t get injured in any of the investigated cases in our work as the localization of the skin incision was drawn laterally and dorsally to the ASIS. This finding corresponds with the landmark of danger zones introduced by Ropars et al., which locates the area of risk for the femoral branch of the LFCN distally and medially to the location of the skin incision used in this work [22]. In return, the gluteal branch got hurt in 50% (left side)/30% (right side) during performance of the current work. This finding also corresponds with the landmark of danger zones introduced by Ropars et al. since our variant of the skin incision is located in the area of risk for the gluteal branch of the LFCN [22]. Both the femoral and the gluteal branch of the LFCN should be considered at risk for injury under use of the anterior approach to the hip joint. This conclusion rests on the results of both Ropars and our work [22]. Injury of one of the two structures has to be considered in a percental range of ca. 40–50%, depending on the mediolateral orientation of the anterior approach. The data we gathered concerning the crossing point of the gluteal branch with the line of skin incision demonstrated that this point was located 28.4 ± 8.62 mm (left side)/15 ± 12.45 mm (right side) distal from the ASIS. The widespread nature of these results is also reflected in the work of Ropars and al. They came to the conclusion that the gluteal branch crossed the ventral edge of the tensor fasciae latae muscle, in an area between 24 and 92 mm distal to the ASIS [22]. These results show the variability of the topographical position of this branch and should be respected under use of the anterior approach to the hip joint. Another aspect considering the topographical situation of the LFCN was highlighted by Carai et al. The position of the LFCN was considered in relation to the Scarpa Fascia. The nerve was localized underneath the fascia in 88.5% of the investigated cases, and it was found above the fascia in 2.7%. No nerve was found in 8.8% of the investigated cases [24]. The LFCN was found in every investigated case of the current work and it was located underneath the Scarpa Fascia in 100% of them. The position of the nerve’s main trunk within the space in between the Scarpa Fascia and the fascia lata can be considered a constant landmark for anatomical orientation in this region from our point of view. It has to be mentioned that there is an existing danger of confusing the LFCN with the ramus cutaneous lateralis of the Iliohypogastric nerve as far as the position of each to the Scarpa Fascia is concerned. The LFCN can be found rather medially located under the Scarpa Fascia, while the lateral cutaneous branch of the iliohypogastric nerve is located rather laterally and subcutaneous above the iliotibial tractus (Figure 1). The limitations of our study meet the criteria inherent to every anatomical study. Our work only deals with the macroscopical capable injury of the LFCN by cutting it while incising the skin. The damage set by retractors and traction during the operating is not measurable in such a setting. The clinical literature reports damage of the LFCN at 15% under the use of the anterior approach to the hip joint [9]. Nevertheless, affecting the nerve did not impair the functional outcome nor the Harris Hip Score in these cases [9]. Other works report no risk for injury of the LFCN under the use of the anterior approach to the hip joint on the other hand, although the authors chose a more medially located variant that raises the risk for injury of the LFCN in accordance to our findings [20]. To conclude, from an anatomical point of view, the position of the skin incision under the use of an anterior approach to the hip joint should be located as laterally and as distally as possible to avoid injury of the LFCN and its following branches. Injury of the nerve’s main trunk can be considered as unlikely. Depending on the orientation of the skin incision in mediolateral direction, either injury of the gluteal branch or injury of the femoral branch has to be considered at a percental range of 30–50%. Nevertheless, identification of the LFCN cannot be recommended due to the high variation of its following branches and the risk of injury that is associated with such a procedure.

## Figures and Tables

**Figure 1 medicina-57-01283-f001:**
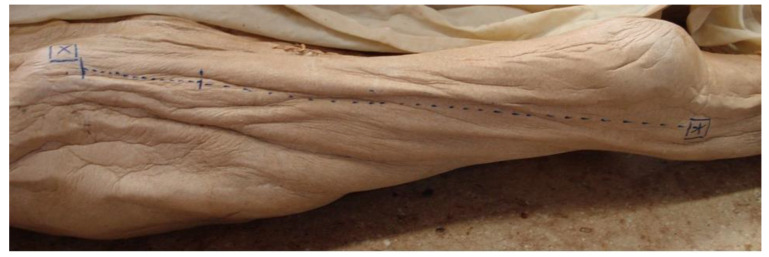
Axis for the orientation of the skin incision. X = ASIS ***** = caput fibulae.

**Figure 2 medicina-57-01283-f002:**
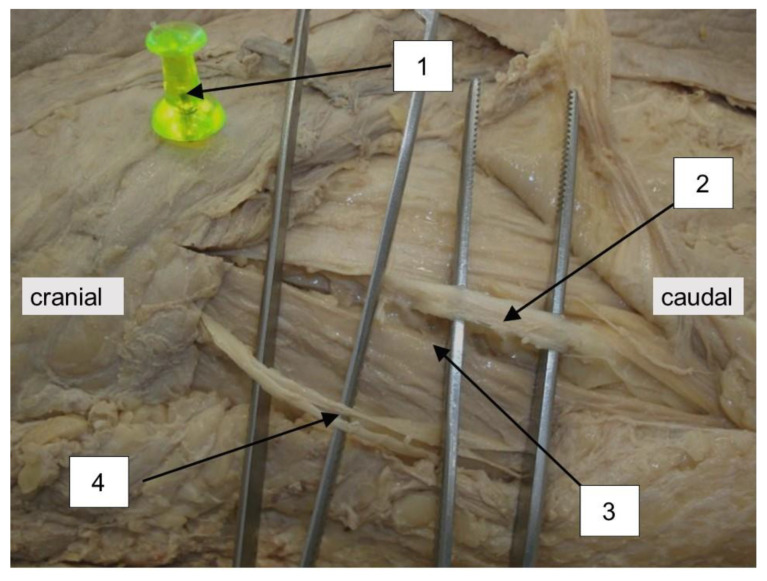
Topographic position of the LFCN to the line of the skin incision of the anterior approach. 1 = ASIS, 2 = LFCN, 3 = line of skin incision, 4 = ramus anterior of the Iliohypogastric Nerve.

**Figure 3 medicina-57-01283-f003:**
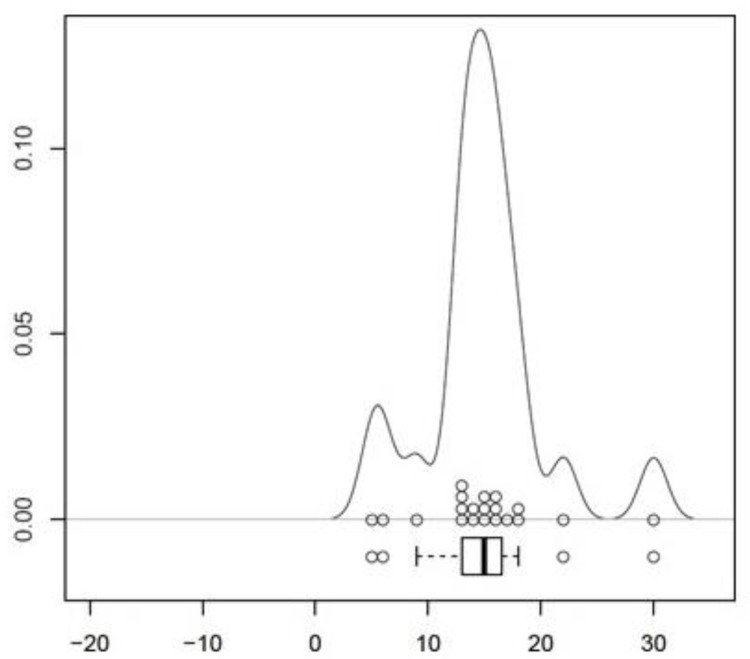
Visualization of the results of measurement on the left side as density-analysis. Scaling in mm. The percental spreading of the single results is displayed on the *y*-axis.

**Figure 4 medicina-57-01283-f004:**
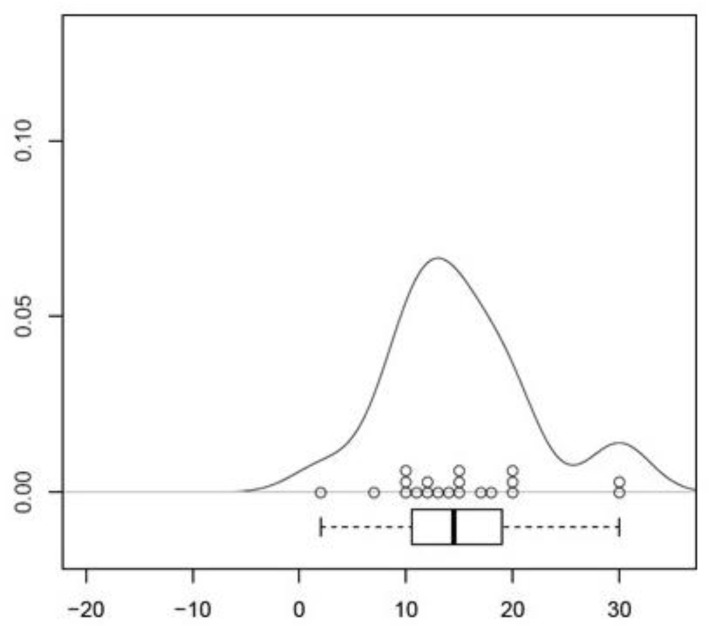
Visualization of the results of measurement on the right side as density-analysis. Scaling in mm. The percental spreading of the single results is displayed on the *y*-axis.

**Figure 5 medicina-57-01283-f005:**
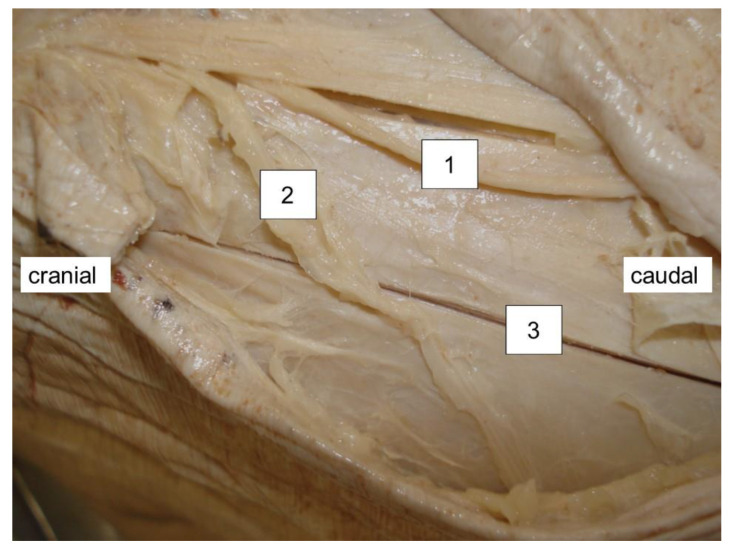
Situation at the branching of the main trunk and the gluteal branch in. relation to the skin incision. 1 = main trunk, 2 = gluteal branch, 3 = skin incision.

**Table 1 medicina-57-01283-t001:** Statistical analysis of the measurements concerning the distance of the nerve’s main trunk to skin incision on the left side.

Mean value	14.90 mm
Standard error	1.19 mm
Confidence interval (CI)	95%, 12.42–17.38 mm
*p*	<0.001
Direction to the skin incision	100% medial (n = 20)

**Table 2 medicina-57-01283-t002:** Statistical analysis of the measurements concerning the distance of the nerve’s main trunk to skin incision on the right side.

Mean value	15.05 mm
Standard error	1.53 mm
Confidence interval (CI)	95%, 11.86–8.24 mm
*p*	<0.001
Direction to the skin incision	100% medial (n = 20)

## Data Availability

The data contained in this article is the authors own work and available according to the terms and conditions of the Creative Commons Attribution (CC BY).

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
