# Peer review of "Minimally Invasive Approaching in Hip Surgery—An Anatomical Investigation of 20 Specimens"

_medicina, 2021, doi:10.3390/medicina57111283_

Round 1

Reviewer 1 Report

The manuscript is suitable for the purpose of the special edition it’s been assigned to

Minor revision include minor English editing

For the rest the paper is ok, it’s an interesting anatomical study on anterior approach in tea

Author Response

Dear reviewers.

Thank you for your time and the effort invested in reviewing our manuscript.

We do believe your suggestions helped improving our work.

Please find a point-by-point answer document in the section below.

The revised manuscript additionally contains track changes for you convenience.

Kind Regards

Clemens Schopper

Rev 1

The manuscript is suitable for the purpose of the special edition it’s been assigned to

Minor revision include minor English editing

For the rest the paper is ok, it’s an interesting anatomical study on anterior approach in tea

Thank you for these valuable comments. Langugage editing was performed.

Reviewer 2 Report

Dear authors, 

I found your artical very interesting for both anatomical scientist and clinicians. I do not have any objection for the  study design, used materials and methods, statistical analysis and presentation of the results. 

But, I have found few errors and suggest minor language changes:

  1. Abstract section

Line 11. typing error – each side of the body….

Line 15. typing error – No injury of the main trunk….

  1. Introduction section

Line 50. – were defined: main trunk, gluteal trunk….

  1. Results section

Line 102. – instead “intersexual compare” I suggest “gender-related differences”

Line 120. - instead “intersexual compare” I suggest “gender-related differences”

Author Response

Dear reviewers.

Thank you for your time and the effort invested in reviewing our manuscript.

We do believe your suggestions helped improving our work.

Please find a point-by-point answer document in the section below.

The revised manuscript additionally contains track changes for you convenience.

Kind Regards

Clemens Schopper

Rev 2

  1. Abstract section

Line 11. typing error – each side of the body….

Line 15. typing error – No injury of the main trunk….

  1. Introduction section

Line 50. – were defined: main trunk, gluteal trunk….

  1. Results section

Line 102. – instead “intersexual compare” I suggest “gender-related differences”

Line 120. - instead “intersexual compare” I suggest “gender-related differences

Thank you for these useful comments. The suggested corrections were performed.

Reviewer 3 Report

Schopper et al described the feasibility of a minimally invasive approach for hip surgery.

The paper is well written and describes the approach adequitly. Nonetheless, I have concerns about the description of the statstical approach and design. for starters the number of cadavers used is not justified through a G-power analysis, second, the authors describe the use of the Shapiro-Wilks test to examine normality which is very good. however, what tests were conducted later on and what was compared is not clear. in Page 4 and in the table I see P-values that are not indicated to compare what with what? if there is a pure descriptive statistic, what does the P-value indicate and where are the remaining values (maximum: minimum, standard deviation, upper and lower limits as well as variance. those would reflect how robust are the data.

The authors are encouraged to address this issue more thoroughly.

Author Response

Dear reviewers.

Thank you for your time and the effort invested in reviewing our manuscript.

We do believe your suggestions helped improving our work.

Please find a point-by-point answer document in the section below.

The revised manuscript additionally contains track changes for you convenience.

Kind Regards

Clemens Schopper

Rev 3

The paper is well written and describes the approach adequitly. Nonetheless, I have concerns about the description of the statstical approach and design. for starters the number of cadavers used is not justified through a G-power analysis, second, the authors describe the use of the Shapiro-Wilks test to examine normality which is very good. however, what tests were conducted later on and what was compared is not clear. in Page 4 and in the table I see P-values that are not indicated to compare what with what? if there is a pure descriptive statistic, what does the P-value indicate and where are the remaining values (maximum: minimum, standard deviation, upper and lower limits as well as variance. those would reflect how robust are the data.

Thank you very much for these valuable suggestions for the improvement of our manuscript. We modified the section Materials & Methods and Results, respectively for this purpose.

This manuscript is a resubmission of an earlier submission. The following is a list of the peer review reports and author responses from that submission.

Round 1

Reviewer 1 Report

The authors evaluated the feasibility for the anterior,  minimally invasive approach to the hip joint in each side of the ten bodies of 20 cadavers (13 females/7 males). The paper is interesting, and some clarifications need to be addressed for publications:

Majors
The statistical analysis focuses on only one parameter, and it seems that distance is not the only criteria for not damaging the nerves.
For instance, the authors mimic only the opening and not a complete surgery or more invasive intervention. Would you please elaborate on how a distance of 15mm is sufficient?
What are the guidelines for the surgeon? Furthermore, please compare your rate to accurate life data if possible.\

Please provide subgroups analyses, between gender (13 to 7 is enough for parametric tests), what are the mean age of the patients?
Did they have the same morphometrical characteristics? 

Would you please provide a general drawing where all the anatomical landmarks needed to understand the papers are presented?

Minors

Merge figure 1 and 4
review the writing and make some paragraphs, some sentences are very long, and the block of text makes the paper very difficult to read. 

Reviewer 2 Report

Thank you for the anatomical description of the minimal invasive anterior approach. There are certainly some clinical implications from the description of your findings. As a surgeon who does not use the anterior approach, I found it hard to follow. The manuscript could benefit from some more clarity and better pictures. I am still unsure where the incision is based on the pictures. I also would like to understand the effect of the section of the gluteal branch. Would a picture be of help? What are the clinical ramifications? I believe you are proponents of the minimal invasive anterior approach and it looks like you are omitting or minimising the negatives of the anterior approach. To proceed further I really would like to see above points addressed as well some stylistic and language improvements.